# Transcriptome Analysis of the Oriental Fruit Fly *Bactrocera dorsalis* Early Embryos

**DOI:** 10.3390/insects11050323

**Published:** 2020-05-23

**Authors:** Wei Peng, Shuning Yu, Alfred M. Handler, Hongyu Zhang

**Affiliations:** 1Key Laboratory of Horticultural Plant Biology (MOE), State Key Laboratory of Agricultural Microbiology, China-Australia Joint Research Centre for Horticultural and Urban Pests, College of Plant Science and Technology, Huazhong Agricultural University, Wuhan 430070, China; pengwei@cjlu.edu.cn (W.P.); shuning_yu@yeah.net (S.Y.); 2College of Life Sciences, China Jiliang University, Hangzhou 310018, China; 3USDA/ARS, Center for Medical, Agricultural and Veterinary Entomology, 1700 SW 23rd Drive, Gainesville, FL 32608, USA; al.handler@usda.gov

**Keywords:** *Bactrocera dorsalis*, embryogenesis, sex-determination, embryonic cellularization, gene expression

## Abstract

The oriental fruit fly, *Bactrocera dorsalis* (Hendel), is one of the most devastating and highly invasive agricultural pests world-wide, resulting in severe economic loss. Thus, it is of great interest to understand the transcriptional changes that occur during the activation of its zygotic genome at the early stages of embryonic development, especially the expression of genes involved in sex determination and the cellularization processes. In this study, we applied Illumina sequencing to identify *B. dorsalis* sex determination genes and early zygotic genes by analyzing transcripts from three early embryonic stages at 0–1, 2–4, and 5–8 h post-oviposition, which include the initiation of sex determination and cellularization. These tests generated 13,489 unigenes with an average length of 2185 bp. In total, 1683, 3201 and 3134 unigenes had significant changes in expression levels at times after oviposition including at 2–4 h versus 0–1 h, 5–8 h versus 0–1 h, and 5–8 h versus 2–4 h, respectively. Clusters of gene orthology (GO) and the Kyoto Encyclopedia of Genes and Genomes (KEGG) annotations were performed throughout embryonic development to better understand the functions of differentially expressed unigenes. We observed that the RNA binding and spliceosome pathways were highly enriched and overrepresented during the early stage of embryogenesis. Additionally, transcripts for 21 sex-determination and three cellularization genes were identified, and expression pattern analysis revealed that the majority of these genes were highly expressed during embryogenesis. This study is the first assembly performed for *B. dorsalis* based on Illumina next-generation sequencing technology during embryogenesis. Our data should contribute significantly to the fundamental understanding of sex determination and early embryogenesis in tephritid fruit flies, and provide gene promoter and effector gene candidates for transgenic pest-management strategies for these economically important species.

## 1. Introduction

The oriental fruit fly, *Bactrocera dorsalis* (Hendel), is one of the most devastating and highly invasive agricultural pests in the world that causes severe economic loss due to damage to over 250 types of fruits and vegetables throughout Southeast Asia and several Pacific Islands [1,2]. As *B. dorsalis* has the characteristics of polyphagy, high reproductive capacity, and adaptability, it is a dreaded invasive species with great potential to invade new geographic areas and host plants [1]. Chemical insecticides are currently considered to be the most effective tool to control fruit flies; however, resistance to commonly applied chemical insecticides in *B. dorsalis* has been increasing [3,4], and it is, thus, urgent to develop new environmentally acceptable methods to control this pest and related tephritid pests.

The primary biological method for tephritid fruit fly control has been the sterile insect techniques (SIT); however, sterile males generated by irradiation or chemosterilants in traditional methods have decreased mating performance, which ultimately reduces population control efficiency [5]. However, genetically-enhanced SIT, that include transgenic genetic male sterility and sexing strains, have shown significant potential for fruit fly management [6,7,8,9,10,11,12]. Based on the identification of key sex-determining genes in insects, it has been possible to obtain male-only progeny by knocking down the female determining genes *transformer* (*tra*) and *transformer-2* (*tra-2*) for male-only populations [13,14,15,16,17,18,19]. In addition, biotechnology-based control strategies have been tested for the survival of male-only populations for sterile release. For example, female-specific dominant lethal constructs use the alternative sex-specific splicing of sex-determining genes for the female-specific expression of a lethal effector gene [20,21,22,23].

The early stages of insect embryonic development include a complex interaction between maternal and zygotic genetic information, in which maternal transcripts degrade while the zygotic genome is activated [24]. In *Drosophila melanogaster*, the degradation of maternal mRNA is regulated by the activity of the Pumilio, Smaug proteins and miRNAs in the developing embryo [25,26,27], while the initial activation of zygotic gene transcription is controlled by the zinc-finger protein Zelda (*zelda*, *zld*) which binds to TAG team sites in the promoter regions of downstream genes, activating their expression [28,29]. Such TAG team promoter sites have been found in sex-determination genes such as *sisterless A* (*sisA*) and cellular blastoderm formation genes such as *serendipity a* (*sry-a*), *nullo*, *bottleneck* (*bnk*) and *slow as molasses* (*slam*) [30,31,32,33,34].

The sex-determination cascade genes are widely conserved in dipteran insects—most prominently, the terminal downstream sex-determining gene, *doublesex* (*dsx*) and its upstream regulatory genes *tra* and *tra-2* [35,36,37,38]. In *D. melanogaster*, a combination of X-chromosome linked signal elements (XSE), namely *sisA*, *scute* (*sc*), *outstretched* (*os*) and *runt* as the primary signal, initiate the female-specific embryonic splicing of *Sxl* that results in the continuing production of functional SXL protein in females [39,40,41]. The sex-specific splicing of *tra* in females is also regulated by SXL, resulting in fully functional TRA protein, while the absence of SXL in males results in truncated non-functional TRA proteins [42,43]. In contrast to in *Drosophila*, *Sxl* transcripts have not been found to be sex-specifically spliced in other dipteran insects in the Tephritidae, Muscidae and Calliphoridae and thus have no apparent role in determining somatic and germline sexual differentiation [13,44,45,46]. In those insects, a Y chromosome-linked male determining factor (M-factor) is the primary determinant of sex differentiation, in which the paternally inherited M factor prevents the maternal activation of the zygotic *tra* positive self-regulatory loop, resulting in the successive male-specific mode of *tra* and *dsx* splicing [47,48,49,50,51].

In the current study, we used Illumina sequencing technology to analyze the early embryonic transcriptome of *B. dorsalis* (0–1, 2–4 h, and 5–8 h post-oviposition) during the key stages for the initiation of sex determination during embryogenesis. Our results, for the first time, provide a broader insight into the expression profiles of sex-determination genes in the *B. dorsalis* early embryo transcriptome. We also present a preliminary scenario of early zygotic gene expression during the maternal-to-zygotic transition (MTZ) in this non-model organism. The early embryonic transcriptome and genome data should assist in the identification of the primary Y-linked M-factor signal and in further elucidating the genetic regulation of sex determination in *B. dorsalis*.

## 2. Materials and Methods

### 2.1. Insect Rearing and Embryo Collection

*B. dorsalis* were reared at the Institute of Horticultural and Urban Entomology at Huazhong Agricultural University (Wuhan, China). Larvae were maintained on a banana diet while adult flies were fed on an artificial diet consisting of yeast extract and sugar. All life stages were cultured at 28 °C under a 12 h light/12 h dark photoperiod in cages [52]. Embryos were collected from females by inducing egg laying through a perforated paper-cup coated with orange juice for a period of 15 min (first egg batch discarded) and then placed on moist filter paper and maintained at 25 °C and 70% humidity for set time intervals.

### 2.2. RNA Extraction and Library Preparation for Transcriptome Sequencing

Embryos were collected at 0–1, 2–4, and 5–8 h after egg laying (AEL) and stored in RNAlater^®^ Solution (Ambion, Austin, TX, USA). Total RNA was extracted using RNAiso Plus reagent (TaKaRa, Dalian, China) according to the manufacturer’s protocol, with the quantity and purity examined using a Bioanalyzer 2100 (Agilent, CA, USA). A total amount of 1 μg RNA from 0–1, 2–4, and 5–8 h embryos was used as the input material to prepare RNA samples. Sequencing libraries were produced using the NEBNext^®^ Ultra™ RNA Library Prep Kit for Illumina^®^ (NEB, Ipswich, MA, USA) according to the manufacturer’s instructions. The index codes were added to sequences to distinguish each sample.

In brief, mRNA was purified from total RNA using poly-T oligo-attached magnetic beads. The divalent cations were used for fragmentation under elevated temperature in NEBNext First Strand Synthesis Reaction Buffer (5X), and the random hexamer primer and M-MuLV Reverse Transcriptase (RNase H^−^) were used for first strand cDNA synthesis. Subsequently, DNA Polymerase I and RNase H were used for second strand cDNA synthesis, after which the exonuclease and polymerase were used to convert the remaining overhangs into blunt ends. Hybridizations were prepared by ligating the NEBNext Adaptor with a hairpin loop structure after the adenylation of the DNA fragment 3′ ends. The library fragments were purified with the AMPure XP system (Beckman Coulter, Beverly, MA, USA) to preferentially select 250–300 bp cDNA fragments. Then, 3 μL of USER Enzyme (NEB, Ipswich, MA, USA) was incubated with the size-selected and adaptor-ligated cDNA at 37 °C for 15 min, followed by 5 min at 95 °C before PCR. The Phusion High-Fidelity DNA polymerase, Universal PCR primers and Index (X) Primer were used to conduct PCR. PCR products were then purified (AMPure XP system), with library quality being assessed using the Agilent Bioanalyzer 2100 system.

The cBot Cluster Generation System was employed to conduct the clustering of index-coded samples using the TruSeq PE Cluster Kit v3-cBot-HS (Illumina, San Diego, CA, USA) according to the manufacturer’s protocol. After cluster generation, an Illumina HiSeq platform was used to sequence the library preparations and to generate 125 bp/150 bp paired-end reads.

### 2.3. Transcript Sequence Analysis

The in-house perl scripts were used first to process raw data (raw reads) of fastq format. After removing reads containing adaptor, reads containing poly-N, and low quality reads from raw data, the clean data (clean reads) were generated. The Q20, Q30 and GC-content of the clean data were calculated simultaneously, and high quality clean data were used for all of the downstream analyses. The paired-end clean reads were then aligned to the whole genome sequence (WGS) of *B. dorsalis* (NCBI Assembly #ASM78921v2) using HISAT2 v2.0.4. Since HISAT2 can generate a database of splice junctions based on the gene model annotation file, it was the preferred mapping tool versus non-splice mapping tools. The read numbers mapped to each gene were counted using HTSeq v0.9.1, and the expression level of each gene was further calculated by FPKM (Fragment Per Kilobase of exon model per million mapped reads) based on the length of the gene and the read counts mapped to this gene. FPKM is presently the most commonly utilized method for assessing gene expression levels as an effect of sequencing depth and gene length for the concurrent read counts [53,54].

### 2.4. Comparison of Differentially Expressed Genes

Prior to differential gene expression analysis, the read counts were adjusted by the edgeR program package through one scaling normalized factor in each sequenced library. Differential expression analysis of two conditions was performed using the DEGSeq R package (1.20.0). The Benjamini and Hochberg’s method was used to adjust the P values. Genes with a corrected P-value less than 0.005 and log_2_(Fold change) of 1 identified by DESeq were assigned as differentially expressed [55]. The GOseq was applied for the Gene Ontology (GO) enrichment analysis of differentially expressed genes, and the gene length bias was corrected. Significantly enriched differentially expressed genes were determined by GO terms with a corrected P-value less than 0.05. Kyoto Encyclopedia of Genes and Genomes (KEGG) is a database resource to check the statistical enrichment of differentially expressed genes using the molecular-level information from biological systems and large-scale molecular datasets from genome sequencing and other high-throughput experimental technologies (http://www.genome.jp/kegg/) [56]. The statistical enrichment of differentially expressed unigenes (DEGs) in KEGG pathways was analyzed with the KOBAS software [57]. Both known and novel transcripts were constructed and identified using the Cufflinks v2.1.1 Reference Annotation Based Transcript (RABT) assembly method from TopHat alignment results [58]. The software rMATS v3.2.5 was used to classify the alternative splicing (AS) events as five basic types. The number of AS events was estimated separately for each compared group.

### 2.5. Quantitative Real-Time PCR

The developmental transcript expression profiles of sex determination and early zygotic genes were investigated using quantitative Real-Time PCR (qPCR). Total RNA was extracted using RNAiso Plus reagent (TaKaRa, Dalian, China) from 100 embryos per replicate, with 200 ng for each sample subjected to reverse transcription for mRNA using the PrimeScript^TM^ RT Master Mix (TaKaRa, Dalian, China). The reverse transcription products were used for qPCR using the primers listed in Appendix A. qPCR was performed using the SYBR Green qPCR mix following the manufacturer’s instructions in a real-time thermal cycler (Bio-Rad, Hercules, CA, USA) using the cycling conditions: 95 °C for 10 min, 40 cycles of 95 °C for 15 s, 60 °C for 30 s and 72 °C for 30 s. Three biological and three technical replicates were performed, with expression data being analyzed by the 2^−ΔΔct^ method [59]. Dissociation curves were determined for each mRNA to confirm unique amplification. The expression of ribosomal protein 49 (*rp49*) was used as an internal control to normalize the expression of mRNA.

### 2.6. Statistical Analysis

Conceptual amino acid sequences encoded by the *tra*, *tra-2* and *dsx* genes in indicated species were used to construct a phylogenetic tree using MEGA5.0 (Molecular Evolutionary Genetics Analysis, Version 5.0, Sudhir Kumar, AZ, USA) with the pair-wise deletion option under the JTT empirical amino acid substitution model. Branch support was assessed by bootstrap analysis with 1000 replicates. All experiments were repeated at least three times and analyzed using GraphPad Prism 5.0 (GraphPad Software, San Diego, CA, USA) or Microsoft Excel (Microsoft, Redmond, WA, USA), with results being expressed as the mean ± SEM. Data were compared with either a two-way ANOVA—with subsequent *t* tests using Bonferroni post-tests for multiple comparisons—or with Student’s *t* test. For all tests, differences were considered significant when *p* < 0.05.

## 3. Results

### 3.1. B. Dorsalis Early Embryo Sequencing and Assembly

Transcriptome libraries (accession number: GSE118472) for the *B. dorsalis* early embryonic developmental stages (0–1, 2–4, and 5–8 h embryos AEL) were constructed by paired-end (PE) Illumina sequencing, which generated a total of 272,036,418 125 bp/150 bp-long PE reads with high sequence quality. The filtered sequence reads from all samples were assembled and produced 13,489 unigenes. The average size of these unigenes was 2185 bp with lengths that ranged from 61 to 57,167 bp (Table 1). There were 9809, 5227, 2873 and 1066 unigenes that were larger than 1000, 2000, 3000 and 5000 bp, respectively, with all having a complete open reading frame (ORF) (Appendix A). After removing low-quality reads, 0–1, 2–4, and 5–8 h libraries generated 83, 85 and 98 million clean reads, respectively. Among these clean reads, 72–84 million (85.12%–86.97%), were mapped to unigenes in the whole genome sequence (WGS) of *B. dorsalis* (Table 2). The percentage of clean reads ranged from 98.08% to 98.61%, and the percentage of reads mapped to the genome exon regions ranged from 91.2% to 97.1%, reflecting high quality sequencing (Appendix A).

### 3.2. Comparison of Gene Expression Profiles in Three Sequential Early Embryo Stages

To evaluate the relative expression level of unigenes in the *B. dorsalis* early embryo transcriptome, unigene paired-end read counts were normalized by transforming them into Fragments Per Kilobase of transcript per Million mapped reads (FPKM). We obtained a wide range of expression levels from less than 1 FPKM to approximately 6595 FPKM (Appendix A). We observed that 47.76%, 47.26%, and 42.38% of the unigenes had a low expression level (FPKM 0–1); 37.96%, 40.40%, and 45.52% had a mid-expression level (FPKM 1–60); and 14.28%, 12.34%, and 12.10% exhibited a high expression level (FPKM > 60) in the 0–1, 2–4 and 5–8 h transcriptome libraries, respectively (Figure 1). The three early embryo stages were evaluated in three pairwise comparisons: 2–4 h vs. 0–1 h, 5–8 h vs. 0–1 h, and 5–8 h vs. 2–4 h, and unigenes found to have significant differences in expression were identified in each comparison (Figure 2). A comparison of the results showed the expression of 1683 unigenes was significantly different between 2–4 and 0–1 h. Of these unigenes, 369 were up-regulated and 1314 were down-regulated (Figure 2a and Appendix A). In the comparison of 5–8 and 0–1 h, the expression profiles of 3201 unigenes had altered, of which 1451 were up-regulated and 1750 were down-regulated (Figure 2b and Appendix A). When comparing 5–8 and 2–4 h, the expression of 3134 unigenes was significantly different as 1675 unigenes were up-regulated and 1459 unigenes were down-regulated (Figure 2c and Appendix A). The expression of 368 unigenes was significantly different in three pairwise comparisons (Figure 3).

### 3.3. Unigene Function Annotation and Analysis

Unigene sequences were annotated by searching the nonredundant NCBI protein database using BLASTX. A total of 11,693 distinct sequences (86.69% of the unigenes) matched known genes (Appendix A). Based on the gene ontology (GO) classification, differently expressed unigenes were characterized within three groups: biological processes, cellular components, and molecular function (Appendix A). The results of each comparison showed high concordance with unigenes related to biological processes (Figure 4). In the comparison of 2–4 and 0–1 h, and 5–8 and 0–1 h, 127 and 421 biological process unigenes were involved in the cellular macromolecule metabolic process (Figure 4a,b). In the comparison of 5–8 and 2–4 h, 607 biological process unigenes were involved in the cellular metabolic process (Figure 4c). In the molecular function category, enrichment comparisons showed that 93 unigenes involved in nucleic acid binding were enriched at 2–4 h vs. 0–1 h, 804 unigenes involved in binding were enriched at 5–8 h vs. 0–1 h, and 496 unigenes involved in organic cyclic and heterocyclic compound binding were enriched t 5–8 h vs. 2–4 h (Figure 4). In addition, the UniGenes metabolic pathway analysis was conducted using the Kyoto Encyclopedia of Genes and Genomes (KEGG) annotation system in each comparison (Appendix A). We found that spliceosome pathway was enriched in the comparison of stages 2–4 and 0–1 h, and 5–8 and 0–1 h (Figure 5). The observation that the RNA binding and spliceosome pathway were highly enriched and overrepresented in the GO term and KEGG analysis during *B. dorsalis* early stage embryogenesis may be an indication that RNA binding and spliceosome pathway enrichment is consistent with sex-specific alternative splicing.

### 3.4. B. dorsalis Sex Determination and Early Zygotic Genes

The alternative intron splicing events that occur during early embryogenesis in *B. dorsalis* include the skipped exons (SE), alternative 5′ splice sites (A5SS), alternative 3′ splice sites (A3SS), mutually exclusive exons (MXE), and retained intron (RI), of which exon skipping exhibited the highest distribution in the three developmental time periods compared (Appendix A). In addition to the *Bdtra*, *Bdtra-2* and *Bddsx* genes, we identified the expression of another 18 additional sex determination gene transcripts in early embryos (Table 3). We found that all sex determination genes contained full length ORFs and that most of the identified sex determination ortholog genes were highly expressed, except for *Bddsx*, where low levels (normalized FPKM < 1) did not change significantly among 0–1, 2–4 and 5–8 h embryos (Table 3). The phylogenetic analysis of the TRA, TRA-2 and DSX amino acid sequences showed that the putative orthologs of *Bdtra*, *Bdtra-2* and *Bddsx* were conserved in tephritid species, especially among the *Bactrocera* (Figure 6). The qPCR results showed that the expression of *Bdtra*, *Bdfru*, *BdSxl* and *Bddpn* increased significantly from 0–1 to 5–8 h, while the expression of *Bddsx*, *Bdtra-2*, *Bdda* and *Bdfl(2)d* had no substantial changes across the three early embryo stages, and the expression of *Bdotu* decreased significantly from 0–1 to 5–8 h (Figure 7). All these results were consistent with our deep sequencing data, which indicated that the current analysis is accurate. The maternal-to-zygotic transition is the period in *B. dorsalis* embryogenesis when maternal transcripts degrade and the zygotic genome expression is activated. In *B. dorsalis*, there were two waves of zygotic transcription at 25 °C, a minor wave occurring early where *BdsisA* and *Bddpn* genes were activated before 4 h after oviposition, and a second major wave that occurred at 5 h when *Bdtra*, *BdSxl* and *Bdfl(2)d* gene transcription was initiated based on the qRT-PCR and mRNA-sequencing results (Figure 8). The waves of zygotic transcription during *B. dorsalis* embryogenesis are similar to those in *C. capitata* [60]; however, *in D. melanogaster*, the minor wave initiates before 2 h after oviposition and the second wave occurs at 3 h, at the time when zygotic *Dmtra* initiates transcription (Figure 8). In the *B. dorsalis* early embryonic transcriptome, we identified a *BdZelda (BdZld)* sequence that encodes a zinc-finger transcription factor protein (Table 3), but we failed to identify *Bdslam* encoding transcripts in the transcriptome. *BdZld* displayed a significant increase in expression from 0–1 to 5–8 h, which was involved in the maternal-to-zygotic transition (Table 3). Other two cellularization genes, *Bdsrya* and *Bdnullo*, were found through BLAST annotation and motif searches of the transcriptome (Table 3). *Bdsrya* showed greater amino acid similarity to the *D. melanogaster srya*-like gene, which is involved in cellular blastoderm formation.

## 4. Discussion

In previous studies, the developmental gene expression profiles of *B. dorsalis* have been investigated by constructing RNA-seq libraries for different developmental stages [61,62,63,64]. However, early embryogenesis transcripts in this dipteran species have been largely unexplored. To better define the broad gene expression profiles in early embryogenesis and the initiation of zygotic control of sex determination, we sequenced transcriptomes from three developmental time periods in *B. dorsalis* early embryos by next-generation sequencing (NGS) technology. Twenty-four sex determination and cellularization genes from a total of 13,489 unigenes were identified, and differential expression profiles of transcripts were validated from the early embryo libraries. Importantly, this study is the first transcriptome exploration of embryogenesis in *B. dorsalis*, which provides an essential gene expression resource necessary to fully understand the genetic basis of sex determination in this tephritid fruit fly.

In *D. melanogaster*, the combination of X-chromosome linked signal elements (XSEs), namely *sisA*, *sc*, *os* and *runt* as the primary signal, initiates the sex determination cascade [41,43,65,66]. In our study, we identified *B. dorsalis* homologs of *sisA*, *sc* and *runt*, but not *os*, expressed in the early embryonic stages. *BdsisA*, *Bdsc* and *Bdrunt* increased 7-, 57- and 45-fold, respectively, over this time period. Compared to the *sisA*, *sc*, *os* and *runt* genes that are all transcribed from the X chromosome in *D. melanogaster* [66,67], *BdsisA*, *Bdsc* and *Bdrunt* are autosomal genes in *B. dorsalis*. The transcripts of *BdsisA* and *Bddpn* have the same expression profiles in *C. capitata*, which exhibit a significant increase in the zygote [60]. Although the primary sex determination signal is postulated to be a Y chromosome-linked male determining factor in tephritids [47,48,49,50,51], the identification of XSEs in the early embryonic developmental transcriptome may expand the function of these genes in the sex determination cascade of non-drosophilid insects [60,68,69,70,71,72,73]. In contrast to in *Drosophila*, *Sxl* is not sex-specifically spliced and is not involved in tephritid sex determination [46,74,75]. In our study, we identified the orthologs of *Drosophila Sxl*, *tra*, *tra-2*, and *dsx* genes in *B. dorsalis*, of which *BdSxl* and *Bdtra-2* have the same transcripts in males and females. *Bdtra* and *Bddsx* play an important role in *B. dorsalis* sex determination and reproduction, which are regulated by sex-specific alternative splicing [17,76,77,78]. Interestingly, we observed that RNA binding, the spliceosome pathway and alternative splicing events were highly enriched during the early stage of embryogenesis in *B. dorsalis*, which is also consistent with the role of sex-specific alternative splicing in *B. dorsalis* sex determination, in addition to that in other insects species [75,79,80]. The expression of *Bdtra* increased approximately two-fold during early embryogenesis, while *Bdtra-2* had consistently higher expression levels during the same periods. Given that TRA-2 performs a novel function by interacting with TRA to form a splicing factor that results in the female-specific splicing and autoregulation of *tra*^F^ and *dsx*^F^ [13,14,15,16,17,18,38,81,82], their significant expression levels in early embryos is not unexpected. However, the relative differences in the levels of *Bdtra-2* transcription had no substantial changes across the three early embryo stages.

Previous research shows that several genes expressed during the minor wave and major wave of embryonic transcription are involved in cellularization and sex determination [24,28,60,83]. Based on FPKM analysis, almost all the genes identified from the *B. dorsalis* early embryos have maternally derived transcripts (Table 3), except for *Bddpn*, *Bddsx*, *Bdfru*, *Bdsc*, *Bdemc*, *Bdrunt*, and *Bdnullo* having FPKM values <1 at 0–1 h. A *Bdslam* transcript was not detected previous to cellularization but has been reported as a maternal transcript in *C. capitata* and *Bactrocera jarvisi* [60,69]. *Bdsrya* was expressed during early embryogenesis at 0–1 h, and cognates of these genes have provided promoters that drive a transgenic tetracycline-suppressible embryonic lethality system, first tested in *D. melanogaster* [10] and then in the pest species *C. capitata*, *Anastrepha suspensa, Anastrepha ludens* and *Lucilia cuprina* [9,11,12,84,85,86]. Notably, for two of these species, *A. ludens* and *L. cuprina*, the use of the *srya* promoter resulted in embryonic lethality but also in maternal female sterility, presumably due to the pre-zygotic expression of the lethal effector in their ovaries that could be suppressed by a short-term tetracycline diet [84,85]. This might also occur in *B. dorsalis*, given the appearance of the pre-zygotic maternal *Bd**srya* transcript at 0–1 h. For *Lucilia*, however, the use of the *nullo* promoter avoided pre-zygotic cell lethality [86], and our results would suggest that the use of the *Bd**nullo* promoter, which initiates expression zygotically at 2–4 h, would also be preferable for an embryo-specific lethality system in *B. dorsalis*. Other uses of embryonic and sex-determination genes for population control or the manipulation of various insects include the engineering of early zygotic gene (EZG) promoters in *Drosophila* for the zygotic-specific expression of antidote genes for the maternal-effect dominant embryonic arrest (*Medea*) gene-drive system [87]. For *Anopheles gambiae* mosquitoes, the CRISPR-Cas9 gene drive targeting of the terminal *dsx* sex determination gene has been used to induce female sterility, resulting in incomplete population suppression [88], and in the silkworm, *Bombyx mori*, a female-specific embryonic lethal system has been developed by the CRISPR-Cas9 targeting of the *tra-2* gene [89]. In our study, we have identified several sex determination and early zygotic genes that should also contribute to the development of novel genetic control strategies for *B. dorsalis* and related pest species.

## 5. Conclusions

In summary, RNA-seq analysis was performed to elucidate or confirm the sex determination and cellularization processes at the molecular level during early embryogenesis in *B. dorsalis*, a major invasive agricultural tephritid pest. Numerous genes displayed significant changes in expression during embryogenesis, especially genes involved in sex determination and the maternal-to-zygotic transcriptional transition. The gene expression patterns of *Bdtra*, *Bdtra-2* and *BdZld*, in particular, are an indication of their important roles in early embryogenesis. Overall, the identification of sex-determination and cellularization genes has demonstrated the value of *B. dorsalis* as a non-typical model for early embryonic development and sexual differentiation. This study should also result in the advancement of the identification and use of gene promoters and sex-specific splicing mechanisms from these genes for transgenic-based improvements in pest management strategies for *B. dorsalis* and related fruit fly species.

## Figures and Tables

**Figure 1 insects-11-00323-f001:**
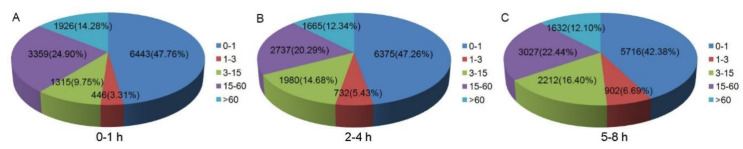
Fragments Per Kilobase of transcript per Million mapped reads (FPKM) distribution of the *B. dorsalis* early embryo transcriptomes.

**Figure 2 insects-11-00323-f002:**
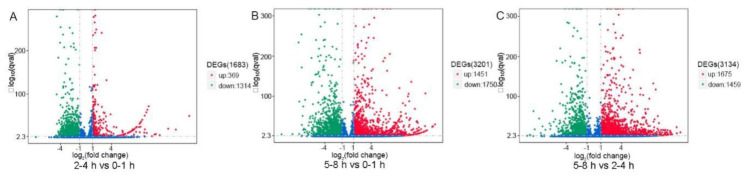
Volcano plot showing differentially expressed unigenes (DEGs, FDR < 0.01 and |log2 ratio| ≥ 1) in 2–4 h vs. 0–1 h, 5–8 h vs. 0–1 h, and 5–8 h vs. 2–4 h libraries of the *B**. dorsalis* early embryos’ transcriptomes. The up-regulated unigenes are represented by a red dot and the down-regulated unigenes, by a green dot.

**Figure 3 insects-11-00323-f003:**
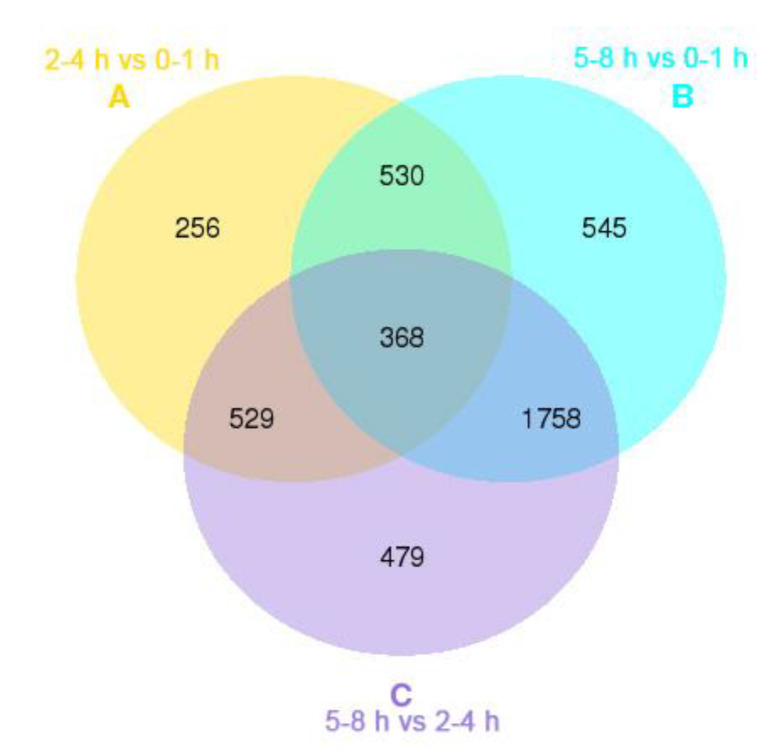
Venn diagram representing the distribution of differentially expressed unigenes in the 0–1, 2–4 and 5–8 h libraries of *B**. dorsalis* early embryos. A, B and C represent the number of differentially expressed unigenes between 2–4 and 0–1 h, between 5–8 and 0–1 h, and between 5–8 and 2–4 h libraries, respectively.

**Figure 4 insects-11-00323-f004:**
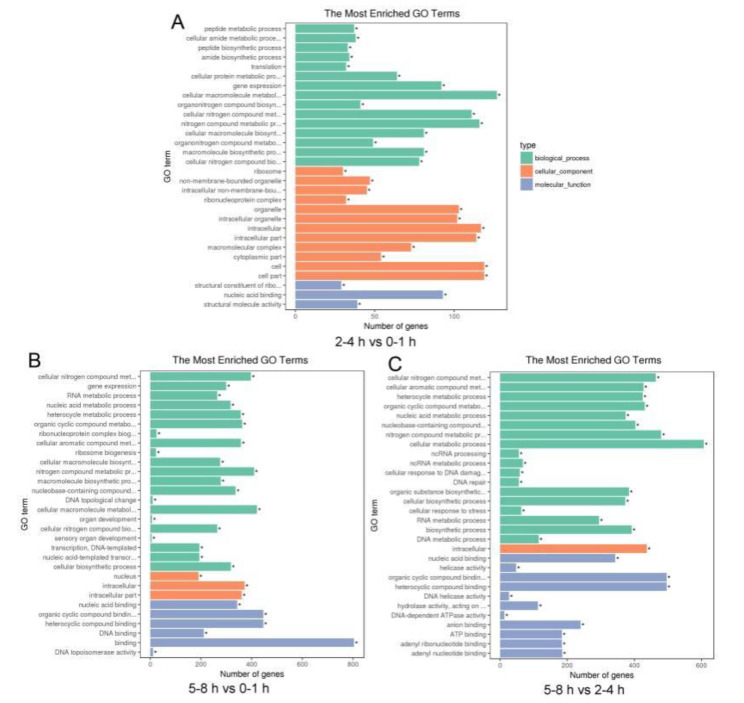
Gene Ontology (GO) terms classification for differentially expressed unigenes (DEGs) during the early embryo transcriptomes of *B**. dorsalis*. (**A**) GO significant enrichment analysis for the DEGs between 2–4 and 0–1 h. (**B**) GO significant enrichment analysis for the DEGs between 5–8 and 0–1 h. (**C**) GO significant enrichment analysis for the DEGs between 5–8 and 2–4 h.

**Figure 5 insects-11-00323-f005:**
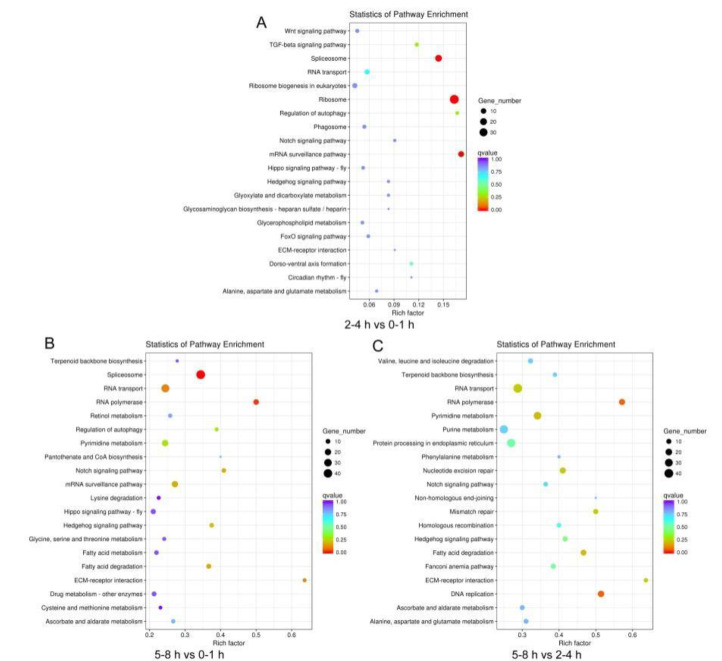
Kyoto Encyclopedia of Genes and Genomes (KEGG) pathway classification for differentially expressed unigenes (DEGs) during the early embryo transcriptomes of *B**. dorsalis*. (**A**) KEGG significant enrichment analysis for the DEGs between 2–4 and 0–1 h. (**B**) KEGG significant enrichment analysis for the DEGs between 5–8 and 0–1 h. (**C**) KEGG significant enrichment analysis for the DEGs between 5–8 and 2–4 h.

**Figure 6 insects-11-00323-f006:**
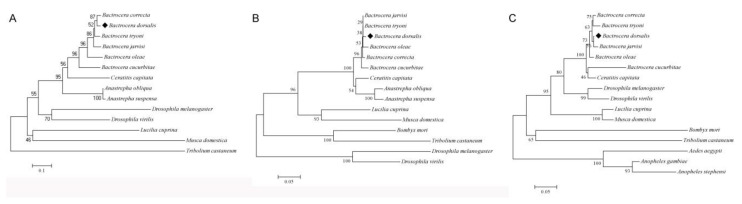
Neighbor-joining tree of insect transformer (TRA), transformer-2 (TRA-2) and doublesex (DSX) amino acid sequences. The scale represents the mean character distance.

**Figure 7 insects-11-00323-f007:**
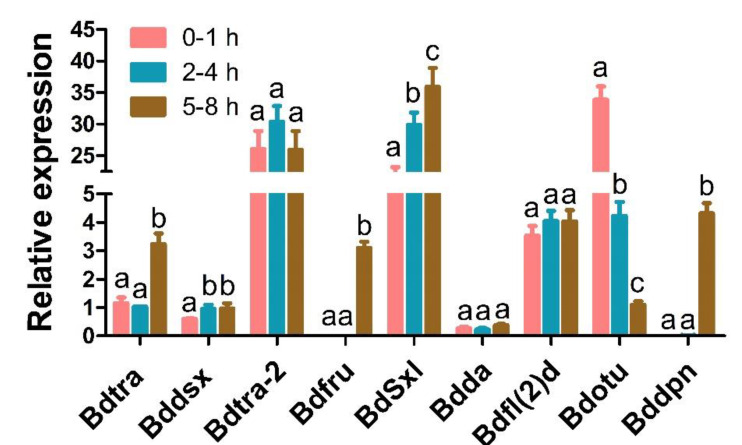
qPCR expression profiles of the sex determination genes during embryogenesis in *B. dorsalis*. Error bars indicate the SEM of three independent biological replicates and different letters above them indicate statistically significant differences (*p* < 0.05) among the three stages based on Student’s *t*-test.

**Figure 8 insects-11-00323-f008:**
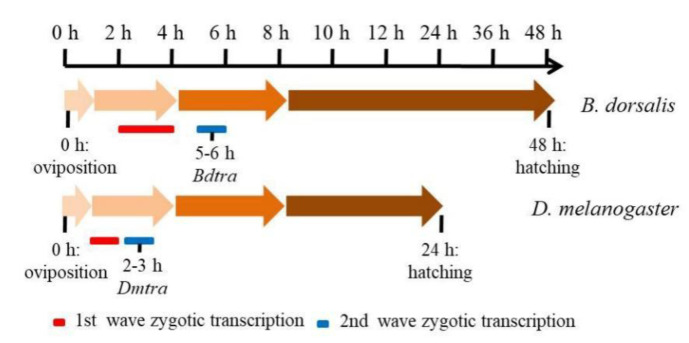
Timing of gene expression during embryogenesis in *B**. dorsalis* and *D. melanogaster*. A schematic representation of the *B**. dorsalis* and *D. melanogaster* embryogenesis stages, from oviposition (0 h) to hatching (48 and 24 h in *B**. dorsalis* and *D. melanogaster*, respectively). The zygotic expression onset of *Bdtra* and the two zygotic expression waves are represented in both species.

**Table 1 insects-11-00323-t001:** Summary of the *B. dorsalis* early embryo transcriptome.

Total Reads	272,036,418
Total number of unigenes	13,489
Average transcript length (bp)	2185
Minimum (bp)	61
Maximum (bp)	57,167
Number of transcripts >1 Kb	9809
Number of transcripts >2 Kb	5227
Number of transcripts >3 Kb	2873
Number of transcripts >5 Kb	1066
Number of transcripts >10 Kb	128

**Table 2 insects-11-00323-t002:** Alignment statistics of the *B. dorsalis* early embryo RNA-Seq analysis.

Sample Name	0–1 h	2–4 h	5–8 h
Raw reads	84,969,396	87,332,604	99,734,418
Clean reads	83,787,930	85,655,698	98,332,254
Clean bases	12.57 G	12.85 G	14.75 G
Total mapped	73,099,585 (87.24%)	73,131,953 (85.38%)	84,393,170 (85.82%)
Multiple mapped	228,708 (0.27%)	223,940 (0.26%)	336,793 (0.34%)
Uniquely mapped	72,870,877 (86.97%)	72,908,013 (85.12%)	84,056,377 (85.48%)
Error rate (%)	0.02	0.02	0.02
Q 20 (%)	97.48	96.93	97.38
Q 30 (%)	92.72	91.65	92.58
GC content (%)	40.94	40.59	40.71

**Table 3 insects-11-00323-t003:** Expression profiles of the sex-determination and cellularization transcripts identified in the *B. dorsalis* early embryo transcriptomes. Expression in each sample is reported as the normalized FPKM value.

Gene Name	Gene ID	Gene Length (bp)	ORF Length (aa)	0–1 h	2–4 h	5–8 h
*Bddaughterless* (*Bdda*)	105221792	3380	710	6.19	4.73	7.30
*Bddeadpan* (*Bddpn*)	105228067	2409	576	0.01	0.59	44.00
*Bddoublesex* (*Bddsx*)	105226634	3858	389	0.25	0.60	0.17
*Bdfemale-lethal(2)d* (*Bdfl(2)d*)	105224548	3326	641	31.72	49.32	51.90
*Bdfruitless* (*Bdfru*)	105224692	3346	839	0	0.03	29.84
*Bdhopscotch* (*Bdhop*)	105225518	3921	1175	62.75	51.02	54.04
*Bdscute* (*Bdsc*)	105227369	1122	296	0	0.10	57.87
*Bdsisterless A* (*BdsisA*)	105231921	1075	259	72.23	164.06	524.97
*BdSex-lethal* (*BdSxl*)	105232012	1938	354	57.67	119.48	169.84
*Bdtransformer* (*Bdtra*)	105232904	2577	422	18.31	19.48	42.76
*Bdtransformer-2* (*Bdtra-2*)	105222300	1676	252	97.08	121.79	91.93
*Bdvirilizer* (*Bdvir*)	105229158	5842	1873	33.11	24.47	47.36
*Bdgroucho* (*Bdgro*)	105227097	2981	727	53.30	99.88	197.99
*BdMes-4*	105222825	4961	1556	77.50	43.23	39.14
*Bdmdg4*	105224476	1808	509	27.18	12.89	33.83
*Bdlongitudinals* (*Bdlo*)	105223842	1964	128	208.09	323.25	216.68
*BdRho1*	105230670	1828	193	207.05	267.43	258.73
*Bdspaghetti-squash* (*Bdsqh*)	105227302	1793	498	43.11	5.09	13.02
*Bdextra-macrochaetae* (*Bdemc*)	105224429	1646	236	0.12	35.34	187.76
*Bdovarian tumor* (*Bdotu*)	105222775	5302	1260	133.30	57.03	17.55
*Bdrunt*	105226934	3210	496	0	69.26	45.79
*BdZelda* (*BdZld*)	105232172	7823	1598	63.76	147.98	120.75
*Bdnullo*	Novel00216	782	211	0	68.06	2.15
*Bdserendipity a* (*Bdsrya*)	105222678	2520	663	32.58	47.88	10.82

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
