# Peer review of "Transcriptome Analysis of the Oriental Fruit Fly Bactrocera dorsalis Early Embryos"

_insects, 2020, doi:10.3390/insects11050323_

Round 1
Reviewer 1 Report
Dr. Peng and collaborators applied NGS technologies to sequence the transcriptome of three different early development stages of Bactrocera dorsalis. They identified a number of differentially expressed genes that relate to RNA binding and spliceosome pathways. They further identify some genes involved in sex determination. Results are discussed with an ample and competent outook at the current literature on Drosophila and other tephritids.
I think this is a good paper, addressing an important issue, with the added value of a significant applied interest. Data analysis includes a variety of descriptive and analytical (one request below) approaches. Supplementary materials adequately includes all the relevant information.
I am listing some minor issues that may nevertheless be addressed before publication:
No replicate has apparently been produced for read counts. At the same time, a thorough statistical analysis has been conducted where read counts are compared across developmental stages using Deseq. In my experience with this package (in a different context) replicates are generally available and obviously contribute stability to the statistical analyses. I am wondering how can Deseq compare read counts without replicates.
I read from the Deseq manual 'Experiments without replicates do not allow for estimation of the dispersion of counts around the expected value for each group, which is critical for differential expression analysis. If an experimental design is supplied which does not contain the necessary degrees of freedom for differential analysis, DESeq will provide a message to the user and follow the strategy outlined in Anders and Huber (2010) under the section 'Working without replicates', wherein all the samples are considered as replicates of a single group for the estimation of dispersion. As noted in the reference above: "Some overestimation of the variance may be expected, which will make that approach conservative." Furthermore, "while one may not want to draw strong conclusions from such an analysis, it may still be useful for exploration and hypothesis generation."
Hence my question: did I miss something in the description of the experimental procedure (i.e. replicates are indeed available), or do the Authors have evidence to support the use of Deseq without replicates? If not, I would urge the Authors to make this clear to the reader and to be very careful/conservative in discussing the results of Deseq differential expression analysis, as suggested by the manual.
Authors apply a relatively non-stringent FDR treshold, would it be worth applying a more conservative approach, especially if the statistical analysis in Deseq is not performing at its best?
Lettering in Figures 1 and 2 is very small, please enlarge.
Use of A,B,C in Figure 3 urges the reader to jump back and forth from the caption to the figure. Please replace with conditions.
Supplementary material reports parimer pairs (qPCR) for more genes than are reported in the text. Please clarify.
Reviewer 2 Report
Wei Peng et al., analyzed transcriptome of early embryos in the oriental fruit fly Bactrocera dorsalis. They applied Illumina sequencing to identify B. dorsalis sex determination genes and early zygotic genes by analyzing transcripts from three early embryonic stages. They found that the RNA binding and spliceosome pathways were highly enriched and overrepresented during the early stage of embryogenesis; sex-determination and cellularisation genes were highly expressed during embryogenesis. Although the transcriptome analysis of the oriental fruit fly (Bactrocera dorsalis) was done in four developmental stages (eggs, third-instar larvae, pupae, and adults) before (https://journals.plos.org/plosone/article?id=10.1371/journal.pone.0029127), the current manuscript provided interesting and meaningful transcriptome data in early embryo developlment. Before publication, I have some minor suggestions.
Line 56 to 59, " In addition, ....for sterile release., this sentence is extremly long and complicated. It needs to be simplified and clear.
Line 161, "200 ng for each sample subjected to reverse transcription", after RNA extraction, did you treat with Dnase to digest genomic DNA, which could affect your qPCR results?
Line 346-347, "However, the basis for the relative differences in their levels of transcription is not apparent." the meaning of this sentence is unclear. It needs to be revised.
